# Magnesium supplementation alleviates drought damage during vegetative stage of soybean plants

Amanda Soares Santos[1], Davielson Silva Pinho[2], Alana Cavalcante da Silva[1], Ramilos Rodrigues de Brito[1], Julian Junio de Jesus Lacerda[1], Everaldo Moreira da Silva[1], Jennyfer Yara Nunes Batista[2], Bruno Sousa Figueiredo da Fonseca[2], Enéas Gomes-Filho[3], Stelamaris de Oliveira Paula-Marinho[1], Alexson Filgueiras Dutra[4], Marcos Renan Lima Leite[5], Alan Mario Zuffo[6], Francisco de Alcântara Neto[7], Jorge González Aguilera[8], José Antonio Rodríguez García[9], Pedro Arias Cubillas[10], Milko Raúl Rivera Campano[9], Alejandro Manuel Ecos Espino[9], Hebert Hernán Soto Gonzales[9], Rafael de Souza Miranda[1]*

1 Postgraduate Program in Agricultural Sciences, Federal University of Piauí, Bom Jesus, Piauí, Brazil,
2 Agronomy Engineering Course, Federal University of Piauí, Bom Jesus, Piauí, Brazil, 3 Postgraduate Program in Biochemistry, Federal University of Ceará, Fortaleza, Ceará, Brazil, 4 Agronomy Engineering Course, Federal Institute of Piauí, Uruçuí, Piauí, Brazil, 5 Postgraduate Program in Agronomy, Federal University of Piauí, Teresina, Piauí, Brazil, 6 Department of Agronomy, State University of Maranhão, Balsas, Maranhão, Brazil, 7 Plant Science Department, Federal University of Piauí, Teresina, Piauí, Brazil, 8 State University of Mato Grosso do Sul, Cassilândia, Mato Grosso do Sul, Brazil, 9 Universidad Nacional de Moquegua, Ilo, Peru, 10 Escuela de Posgrado-Doctorado en Ciencias Ambientales, Universidad Nacional Jorge Basadre Grohmann, Tacna, Peru

☉ These authors contributed equally to this work.
‡ AFD, MRLL, AMZ, FAN, JGA, JARG, PAC, MRRC, AMEE and HHSG also contributed equally to this work.
* rsmiranda@ufpi.edu.br

**Data Availability Statement:** All relevant data are within the manuscript and its Supporting information files.

## Abstract

Our working hypothesis was that magnesium (Mg) supplementation modulates plant performance under low water availability and improves drought tolerance in soybean genotypes. Plants of Bônus 8579, M8808 and TMG1180 genotypes were grown under field conditions and subjected to three water stress treatments (control, moderate and severe stress) and three Mg levels [0.9 (low), 1.3 (adequate) and 1.7 cmolc dm$^{-3}$ (supplementation)]. After 28 days of drought imposition, the growth parameters, osmotic potential, relative water content, leaf succulence, Mg content and photosynthetic pigments were assessed. In general, drought drastically decreased the growth in all genotypes, and the reductions were intensified from moderate to severe stress. Under adequate Mg supply, TMG1180 was the most drought-tolerant genotype among the soybean plants, but Mg supplementation did not improve its tolerance. Conversely, although the M8808 genotype displayed inexpressive responses to drought under adequate Mg, the Mg-supplemented plants were found to have surprisingly better growth performance under stress compared to Bônus 8579 and TMG1180, irrespective of drought regime. The improved growth of high Mg-treated M8808-stressed plants correlated with low osmotic potential and increased relative water content, as well as shoot Mg accumulation, resulting in increased photosynthetic pigments and culminating in the highest drought tolerance. The results clearly indicate that Mg

**Funding:** This study was supported by the Fundação de Amparo à Pesquisa do Estado do Piauí (FAPEPI) under grant Edital PPP FAPEPI/MCT/CNPq/CT-INFRA n° 007/2018, and by the Conselho Nacional de Desenvolvimento Científico e Tecnológico (CNPq) under grant number 427219/2018-3. Additionally, A.S. Santos received a graduate fellowship from the Coordenação de Aperfeiçoamento de Pessoal de Nível Superior (CAPES). The funders had no role in study design, data collection and analysis, decision to publish, or preparation of the manuscript.

**Competing interests:** The authors have declared that no competing interests exist.

**Abbreviations:** ANOVA, analysis of variance; Chl *a*, chlorophyll *a*; Chl *b*, chlorophyll *b*; Chl *total*, chlorophyll *total*; Cs, solute concentration; DM, dry mass; ETc, crop evapotranspiration; ETo, reference evapotranspiration; FM, fresh mass; LA, leaf area; LS, leaf succulence; Mg, magnesium; NL, number of leaves; PCA, principal component analysis; PH, plant height; PSI, photosystem I; PSII, photosystem II; R, universal gas constant; RWC, relative water content; SD, stem diameter; SDM, shoot dry mass; SFM, shoot fresh mass; T, temperature; TM, turgidity mass; Ψs, osmotic potential.

supplementation is a potential tool for alleviating water stress in M8808 soybean plants. Our findings suggest that the enhanced Mg-induced plant acclimation resulted from increased water content in plant tissues and strategic regulation of Mg content and photosynthetic pigments.

## Introduction

Soybean (*Glicine max* L. Merrill) is an important crop explored worldwide. In recent years, Brazil has experienced remarkable growth in agricultural land utilization and has emerged as the world's leading producer of soybean grains [1, 2]. However, environmental constraints such as drought are projected to cause a reduction of up to 40% in soybean yields [3]. A similar situation has also been reported in some regions of other potential soybean producers. In United States and China, drought-induced yield losses of up to 25.4% and 21.8% have been observed over the past four and five decades, respectively [4, 5]. Thus, water limitation has raised as a serious agricultural issue that exerts a substantial impact on grain and food production in arid and semiarid regions [6].

In plants, drought disturbs several physiological and biochemical processes, such as water transport and photosynthetic machinery efficiency, drastically impairing growth parameters such as leaf area and dry mass production [7, 8]. To cope with drought, plants may activate several mechanisms, highlighting morphological, physiological, and biochemical adjustments. The majority of plant species alleviate deleterious drought effects by increasing root growth to provide greater exploration of soil and water uptake [9]; increase photosynthetic capacity by increasing photosynthetic pigment content [10, 11] and electron transport from PSII to PSI [12]; reduce stomatal opening to prevent excessive water loss through transpiration [13], and/or activate pathways for osmotic adjustment, including the accumulation of organic solutes to maintain the water potential gradient and water uptake [14, 15]. The combination of these adjustments can alleviate the damage of water restriction and allow better performance under stressful conditions.

Recent reports have shown that magnesium (Mg) supply not only increase plant growth [16] but also may improve the water efficiency use [17] and activate mechanisms for tolerance to biotic and abiotic stresses [18]. In tomato plants, Mg supplementation activated the antioxidant system and induced systemic resistance against *Ralstonia solanacearum* [19]. In wheat and maize, magnesium was found to improve the performance of photosynthetic machinery and antioxidant enzymes, inducing relevant responses against heat stress [20]. Additionally, Mg supplementation counteracts the deleterious effects of salt stress during the reproductive stage in *Plantago crassifolia* plants [21]. However, evidence into how Mg nutrition modulates the soybean responses to water restriction remains to be explored.

Our working hypothesis is that Mg supplementation triggers water stress tolerance in *G. max* through activation of mechanisms for maintenance of water content in plant tissues. This hypothesis was assessed by analyzing the impacts of three Mg supplies on drought-contrasting soybean genotypes subjected to three water stress levels. Growth parameters and several physiological stress indicators were investigated to elucidate the relationship between magnesium supplementation and water stress tolerance.

## Material and methods

### Experimental conditions and treatments

Experiments were carried out under field conditions at Federal University of Piauí, city of Bom Jesus, Piauí State, with geographic coordinates latitude 9°04'45.6" S and longitude 44°

19'37,9" W, at approximately 277 m altitude. No permits of an authority were necessary to access the field site. The climate of the region is classified as BSh hot semiarid, with summer rains and dry winters according to the Köppen classification, as described in Edvan et al. [22]. The environmental conditions during the experiments are detailed in S1 Fig.

The experiment was set in completely randomized blocks with four repetitions in a $3 \times 3 \times 2$ factorial scheme, corresponding to three soybean genotypes (BÔNUS 8579, M8808 IPRO and TMG 1180), three drought treatments (control—60% crop evapotranspiration (ETc), moderate drought—45% ETc, and severe drought—30% ETc) and two Mg doses [1.3 (adequate) and 1.7 $cmol_c$ $dm^{-3}$ de Mg (supplementation)].

Drought treatments were applied soon after the soybean plants achieved the V4 stage, 28 days after sowing. The chemical properties of the soil at a depth of 0.00–0.20 m were determined before and after the experiment (S1 and S2 Tables). Mg doses were applied during soil correction and fertilization by using different combinations of dolomitic and calcite limestone. Seeds from soybean genotypes were sown through direct planting, with an average plant density of 220,000 plants per ha.

## Irrigation management

Irrigation management was carried out by a drip irrigation system, with drip tubes distributed at 0.50 m and 0.20 m between drippers. The irrigation levels (60, 45 and 30% ETc) were calculated by using the reference evapotranspiration (ETo) calculated by the Penman-Monteith-FAO method, using data from the meteorological station located in the experimental area. The evapotranspiration of the crop (ETc) was calculated according to kc information described in Doorenbos and Kassam [23]. During the imposition of water treatments, the daily averages for irrigation levels consisted of 1.97 mm, 1.60 mm and 1.24 mm for treatments 60, 45 and 30% ETc, respectively.

## Plant harvest and growth analyzes

The plant material was harvested at 28 days after the imposition of drought treatments, before the reproductive stage. A group of plants was used for physiological analysis by using the first fully expanded leaves of two plants, corresponding to one repetition. Another two plants per repetition were harvested and employed to measure the plant growth parameters: plant height (PH), stem diameter (SD), number of leaves (NL), leaf area (LA), shoot fresh mass (SFM, leaves + stems) and shoot dry mass (SDM, leaves + stems).

## Relative water content and leaf succulence

The relative water content (RWC) was measured by weighing leaf discs of 1.0 cm diameter to obtain fresh mass (FM). The discs were emerged in distilled water to obtain the turgidity mass (TM). Finally, the plant material was dried at 65°C for 72 h, and the dry mass was determined (DM) and used to estimate the RWC using the formula [24]:

$$RWC\ (\%) = \frac{(FM - DM)}{(TM - DM)} \times 100$$

Leaf succulence (LS—g $H_2O$ $m^{-2}$) was determined through the formula:

$$LS = \frac{(SFM - SDM)}{LA}$$

## Osmotic potential

The leaf osmotic potential ($\Psi$s) was determined after extraction of cell sap by pressuring leaf tissues. The osmolarity of the cell sap was measured through a vapor pressure micro-osmometer (Model 5600; Vapor, Wescor, Utah, USA). The $\Psi$s values were obtained using Van't Hoff's equation: $\Psi s = -R \times T \times Cs$, being R the universal gas constant (0.08205 l atm $mol^{-1}$ $K^{-1}$), T the temperature (T = 298°K), and Cs the solute concentration (M), typically expressed in atmospheres and converted to MPa.

## Relative tolerance to drought

The relative drought tolerance was measured as the SDM of plants grown in drought conditions related to the SDM of plants in the control treatment.

## Content of Mg

Leaves and stem samples were dried and ground into fine powders and digested in 65% $HNO_3$. Then, the Mg concentrations were estimated by atomic absorption spectrometry according to Malavolta, Vitti and Oliveira [25].

## Photosynthetic pigments

Photosynthetic pigments, chlorophyll *a* (Chl *a*), chlorophyll *b* (Chl *b*), Chl *total* and carotenoids, were extracted from leaf discs using a solution of dimethylsulfoxide (DMSO) saturated with $CaCO_3$ in the dark. The extracts were subjected to absorbance readings at 480, 649 and 665 nm, and concentrations were calculated using equations based on specific absorption coefficients, according to Wellburn [26].

## Statistical analysis

Data were subjected to analysis of variance (ANOVA) to assess the effects of the factors (S3 Table). The treatments were compared by Tukey's test at 0.05 (P < 0.05). A multivariate analysis by the principal component method (PCA) was performed using the SPSS Statistics program.

## Results

The data obtained are presented through clustering analyses and PCA, providing an integrated view. In summary, Figs 1–3 show the comparison for independent factors, namely drought, Mg nutrition, and soybean genotype, respectively. Each figure is accompanied by a specific reference point. Fig 1 illustrates the comparison of drought treatments within different Mg levels and soybean genotypes, taking the data of well-irrigated plants as the reference. Fig 2 displays the results of Mg treatments comparison within drought levels and soybean genotypes, using plants grown under recommended Mg as the reference point. Finally, Fig 3 presents a comparison of soybean genotypes within Mg and drought treatments, with data from the Bônus 8579 genotype serving as the reference. The absolute values and corresponding statistical results are detailed in Tables 1–3 and S3 Table.

## Plant growth performance

The accumulation of biomass in soybean genotypes was differentially regulated by both drought stress and Mg treatments. In general, moderate and severe drought dramatically decreased the plant height (PH), stem diameter (SD), number of leaves (NL), leaf area (LA),

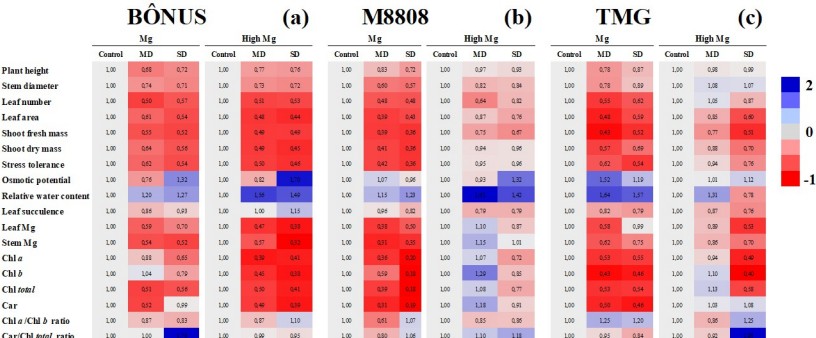

**Fig 1. Results of drought treatments comparison within Mg levels and soybean genotypes.** Clustering analysis of growth and physiological and biochemical indexes relative to changes due to drought treatments in Bônus 8579 (**a**), M8808 IPRO (**b**) and TMG 1180 (**c**) soybean genotypes. The assays were performed in plants 28 days after grown under adequate (Mg) and (High Mg) supplementation magnesium regimes, and subjected to three drought treatments (control, moderate drought—MD and severe drought—SD). Each row represents an individual assay. In all cases, blue color indicates an increase, and red indicates a decrease in the analyzed indexes, taking control plants as reference. Gray indicates no change. Number inside the box and different red and blue intensities express the extent of the change according to fold increase/decrease related to reference. For absolute values and statistical details, see Tables 1–3.

shoot fresh mass (SFM) and shoot dry mass (SDM) of Bônus 8579 plants from adequate (1.3 cmol_c dm$^{-3}$) and supplementation (1.7 cmol_c dm$^{-3}$) Mg treatments compared to the respective controls (Fig 1a and Table 1). Similar results were observed for M8808 IPRO plants under adequate Mg supply, which displayed strong reductions by drought stress in all analyzed growth parameters (Fig 1b); however, under Mg supplementation, drought-induced significant negative impacts were noticed only for NL in moderately stressed M8808 plants (Fig 1b). Conversely, TMG 1180 plants grown in Mg-adequate conditions showed significant decreases under moderate drought only in PH, NL and SFM, whereas Mg- supplemented plants displayed strong reductions in LA, SFM and SDM under severe drought (Fig 1c).

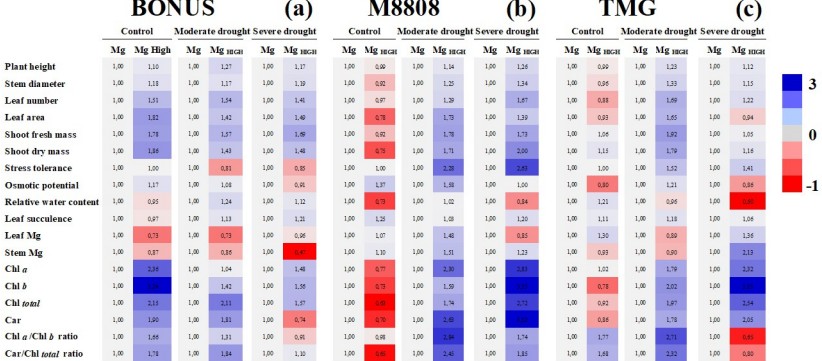

**Fig 2. Results of Mg treatments comparison within drought levels and soybean genotypes.** Clustering analysis of growth and physiological and biochemical indexes relative to changes due magnesium (Mg) treatments in Bônus 8579 (**a**), M8808 IPRO (**b**) and TMG 1180 (**c**) soybean genotypes. The analyzes were carried out in plants after 28 days grown well irrigated (control) and subjected to moderate and severe drought. Each row represents an individual assay. For all cases, the color intensity refers to changes due to Mg supplementation within the soybean genotype and drought treatment. Blue color indicates an increase, and red indicates a decrease in analyzed indexes, taking data from the Mg-grown plants as reference. Number inside the box and different red and blue intensities express the extent of the change according to fold increase/decrease related to reference. For absolute values and statistical details, see Tables 1–3.

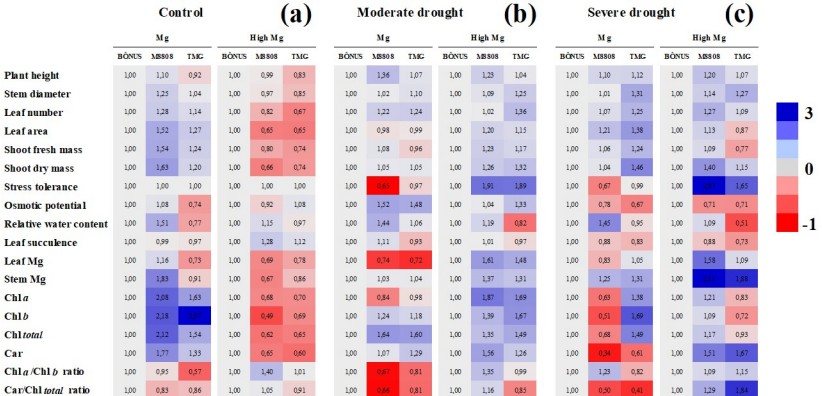

**Fig 3. Results of soybean genotypes comparison within Mg and drought treatments.** Clustering analysis of growth and physiological and biochemical indexes relative to changes due to soybean genotypes. The data were analyzed in plants grown well irrigated (control, **a**) and under moderate (**b**) and severe (**c**) drought, in different magnesium regimes (adequate and supplementation Mg). Each row represents an individual assay. For all cases, the color intensity refers to changes due to the soybean genotype (Bônus 8579, M8808 IPRO and TMG 1180) within the Mg dose and drought treatment. Blue color indicates an increase, and red indicates a decrease in analyzed indexes, taking data from the Bônus 8579 genotype as reference plants. Number inside the box and different red and blue intensities express the extent of the change according to fold increase/decrease related to reference. For absolute values and statistical details, see Tables 1–3.

**Table 1. Mean comparison data for plant height (PH), stem diameter (SD), number of leaves (NL), leaf area (LA), shoot fresh mass (SFM, leaves + stems) and shoot dry mass (SDM, leaves + stems) from soybean genotypes (Bônus 8579, M8808 IPRO and TMG 1180) after 28 days grown under adequate and high magnesium regimes, and subjected to three drought treatments (control, moderate and severe drought).**

| Treatments | | | PH | SD | NL | LA | SFM | SDM |
|---|---|---|---|---|---|---|---|---|
| Stress level | Mg | Genotype | (cm plant$^{-1}$) | (cm plant$^{-1}$) | (leaf plant$^{-1}$) | (cm$^2$ plant$^{-1}$) | (g plant$^{-1}$) | (g plant$^{-1}$) |
| Control | Adequate | Bônus | 39.6±0.9$^{abA}$ | 5.3±0.1$^{bA}$ | 15.6±0.4$^{aA}$ | 1453.7±79.5$^{bA}$ | 48.7±0.1$^{bA}$ | 12.2±0.2$^{bA}$ |
| | | M8808 | 43.8±0.6$^{aA}$ | 6.7±0.4$^{aA}$ | 20.0±1.0$^{aA}$ | 2214.9±147.8$^{aA}$ | 75.1±3.8$^{aA}$ | 19.9±1.4$^{*aA}$ |
| | | TMG | 36.6±0.9$^{bA}$ | 5.6±0.2$^{abA}$ | 17.9±2.9$^{aA}$ | 1848.9±29.2$^{abA}$ | 60.3±6.7$^{bA}$ | 14.6±2.4$^{1bA}$ |
| | High | Bônus | 43.8±2.4$^{aA}$ | 6.3±0.3$^{*aA}$ | 23.6±1.6$^{*aA}$ | 2646.8±2.9$^{*aA}$ | 87.0±6.4$^{*aA}$ | 22.8±1.7$^{*aA}$ |
| | | M8808 | 43.1±0.8$^{aA}$ | 6.1±0.5$^{aA}$ | 19.3±0.6$^{abA}$ | 1731.1±75.0$^{bA}$ | 69.2±4.8$^{bA}$ | 15.0±1.0$^{bA}$ |
| | | TMG | 36.1±1.1$^{bA}$ | 5.4±0.5$^{aA}$ | 15.8±2.9$^{bA}$ | 1712.9±275.8$^{bA}$ | 64.0±11.2$^{bA}$ | 16.8±3.1$^{bA}$ |
| Moderate drought | Adequate | Bônus | 26.8±0.9$^{bB}$ | 4.0±0.2$^{aB}$ | 7.9±0.4$^{aB}$ | 890.5±83.4$^{aB}$ | 26.9±1.3$^{aB}$ | 7.9±0.4$^{aAB}$ |
| | | M8808 | 36.4±0.5$^{aB}$ | 4.0±0.1$^{aA}$ | 9.6±0.7$^{aB}$ | 873.6±58.0$^{aB}$ | 29.1±0.8$^{aB}$ | 8.2±0.6$^{aB}$ |
| | | TMG | 28.8±1.1$^{bB}$ | 4.3±0.1$^{aB}$ | 9.8±0.3$^{aB}$ | 881.4±53.5$^{aB}$ | 25.8±1.3$^{aB}$ | 8.3±0.2$^{aB}$ |
| | High | Bônus | 33.9±0.6$^{*bB}$ | 4.6±0.3$^{bB}$ | 12.2±0.4$^{*aB}$ | 1265.0±48.0$^{aB}$ | 42.2±0.4$^{*aB}$ | 11.2±0.6$^{aB}$ |
| | | M8808 | 41.6±1.1$^{*aA}$ | 5.0±0.1$^{*abA}$ | 12.4±0.7$^{aB}$ | 1513.1±38.1$^{*aA}$ | 51.9±1.5$^{*aB}$ | 14.1±0.4$^{*aA}$ |
| | | TMG | 35.4±0.2$^{*bA}$ | 5.8±0.2$^{*aA}$ | 16.5±1.1$^{*aA}$ | 1451.3±98.7$^{1*aAB}$ | 49.5±3.0$^{*aB}$ | 14.8±1.2$^{1*AB}$ |
| Severe drought | Adequate | Bônus | 28.6±0.2$^{aB}$ | 3.8±0.1$^{bB}$ | 8.9±0.5$^{aB}$ | 786.6±43.6$^{aB}$ | 25.3±0.8$^{aB}$ | 6.9±0.2$^{aB}$ |
| | | M8808 | 31.6±0.5$^{aB}$ | 3.8±0.1$^{bA}$ | 9.5±0.2$^{aB}$ | 953.1±1.9$^{aB}$ | 26.9±1.0$^{aB}$ | 7.2±0.5$^{aB}$ |
| | | TMG | 32.0±0.9$^{aAB}$ | 5.0±0.3$^{aAB}$ | 11.1±0.5$^{aB}$ | 1087.7±23.9$^{aB}$ | 31.4±1.6$^{aB}$ | 10.1±0.6$^{aB}$ |
| | High | Bônus | 33.4±1.1$^{*bB}$ | 4.5±0.3$^{bB}$ | 12.5±1.5$^{*aB}$ | 1173.8±96.8$^{aB}$ | 42.7±1.2$^{*aB}$ | 10.2±1.2$^{aB}$ |
| | | M8808 | 40.0±0.4$^{*aA}$ | 5.1±0.3$^{*abA}$ | 15.9±0.9$^{*aAB}$ | 1323.1±136.1$^{aA}$ | 46.4±2.3$^{*aB}$ | 14.3±1.1$^{*aA}$ |
| | | TMG | **35.9±1.0**$^{abA}$ | **5.7±0.4**$^{aA}$ | **13.6±1.6**$^{aA}$ | **1021.9±66.2**$^{aB}$ | **32.9±1.9**$^{aC}$ | **11.7±1.5**$^{aB}$ |

*Note*: Different capital letters indicate significant differences due to drought stress within the same genotype and Mg level. Different lowercase letters denote significant alterations among soybean genotypes within the same Mg and stress level. The presence of asterisks (*) represent significant differences due to Mg treatment in the same genotype and drought level, according to Tukey's test (p < 0.05).

**Table 2. Mean comparison data for relative drought tolerance, osmotic potential (Ψs), relative water content (RWC) and leaf succulence (LS) from soybean genotypes (Bônus 8579, M8808 IPRO and TMG 1180) after 28 days grown under adequate and high magnesium regimes, and subjected to three drought treatments (control, moderate and severe drought).**

| Treatments | | | Relative tolerance | Ψs | RWC | LS |
|---|---|---|---|---|---|---|
| Stress level | Mg | Genotype | (%) | (MPa) | (%) | (g H$_2$O m$^{-2}$) |
| Control | Adequate | Bônus | 100.0±0.0$^{aA}$ | -1.2±0.2$^{aAB}$ | 18.4±0.0$^{bA}$ | 253.5±13.6$^{aA}$ |
| | | M8808 | 100.0±0.0$^{aA}$ | -1.1±0.2$^{aA}$ | 27.8±0.0$^{*aA}$ | 251.2±15.0$^{aA}$ |
| | | TMG | 100.0±0.0$^{aA}$ | -1.6±0.0$^{aA}$ | 14.2±0.0$^{bB}$ | 246.6±21.3$^{aA}$ |
| | High | Bônus | 100.0±0.0$^{aA}$ | -1.4±0.1$^{aB}$ | 17.6±0.0$^{aB}$ | 245.5±9.8$^{bA}$ |
| | | M8808 | 100.0±0.0$^{aA}$ | -1.5±0.0$^{aA}$ | 20.3±0.0$^{aB}$ | 314.0±28.0$^{*aA}$ |
| | | TMG | 100.0±0.0$^{aA}$ | -1.3±0.3$^{aA}$ | 17.1±0.0$^{aAB}$ | 274.5±13.2$^{abA}$ |
| Moderate drought | Adequate | Bônus | 61.7±1.7$^{aB}$ | -1.6±0.1$^{aB}$ | 22.0±0.0$^{bA}$ | 218.0±18.9$^{aA}$ |
| | | M8808 | 42.0±4.0$^{aB}$ | -1.1±0.1$^{*aA}$ | 31.8±0.0$^{aA}$ | 241.8±18.1$^{aA}$ |
| | | TMG | 62.1±11.2$^{aB}$ | -1.1±0.1$^{aA}$ | 23.3±0.0$^{bA}$ | 202.5±23.8$^{aA}$ |
| | High | Bônus | 49.9±4.0$^{aB}$ | -1.7±0.2$^{aB}$ | 27.4±0.0$^{abA}$ | 246.6±14.7$^{aA}$ |
| | | M8808 | 95.5±6.5$^{*aA}$ | -1.7±0.0$^{aA}$ | 32.5±0.0$^{aA}$ | 249.6±7.1$^{aB}$ |
| | | TMG | 94.5±12.3$^{*aA}$ | -1.3±0.1$^{aA}$ | 22.5±0.0$^{bA}$ | 239.8±8.5$^{aB}$ |
| Severe drought | Adequate | Bônus | 54.5±1.6$^{aB}$ | -0.9±0.1$^{aA}$ | 23.4±0.0$^{bA}$ | 234.8±9.4$^{aA}$ |
| | | M8808 | 36.3±2.4$^{aB}$ | -1.2±0.1$^{aA}$ | 34.0±0.0$^{aA}$ | 206.6±10.8$^{aA}$ |
| | | TMG | 54.2±3.8$^{aB}$ | -1.4±0.1$^{aA}$ | 22.3±0.0$^{*bA}$ | 195.5±10.8$^{aA}$ |
| | High | Bônus | 46.2±7.4$^{bB}$ | -0.8±0.3$^{aA}$ | 26.2±0.0$^{aA}$ | 283.3±28.4$^{aA}$ |
| | | M8808 | 95.6±3.5$^{*aA}$ | -1.2±0.1$^{aA}$ | 28.7±0.0$^{aA}$ | 248.8±25.2$^{abB}$ |
| | | TMG | 76.2±13.3$^{aA}$ | -1.2±0.1$^{aA}$ | 13.3±0.0$^{bB}$ | 207.8±5.4$^{bB}$ |

*Note*: Different capital letters indicate significant differences due to drought stress within the same genotype and Mg level. Different lowercase letters denote significant alterations among soybean genotypes within the same Mg and stress level. The presence of asterisks (*) represent significant differences due to Mg treatment in the same genotype and drought level, according to Tukey's test (p < 0.05).

Significant alterations by Mg treatments in plant growth varied with soybean genotype. For Bônus 8579, Mg-high benefits effects were registered only in plants subjected to moderate drought, where Mg supplementation significantly increased the PH, SFM and SDM in comparison to adequate Mg supply (Fig 2a and Table 1). Mg supplementation seemed to be effective in alleviating the deleterious effects of drought on the M8808 genotype, irrespective of stress level, improving the LA, SFM and SDM in moderately stressed plants and the NL, SFM and SDM in severe drought-stressed plants in comparison to the respective controls (Fig 2b).

Fig 3 presents the normalized data comparison between Bônus 8579, M8808 and TMG 1180 plants within Mg levels and drought treatments. In general, well-irrigated plants from the M8808 and TMG 1180 genotypes showed PH, SD, NL, LA, SFM and SDM values lower than those from Bônus 8579 plants under Mg supplementation, while the opposite occurred for Mg-recommended (Fig 3a and Table 1). Similar growth was registered among soybean genotypes subjected to moderate and severe drought, irrespective of Mg supply (Fig 3b, 3c and Table 1).

## Drought relative tolerance

As a consequence of growth regulation, soybean genotypes exhibited differential relative tolerance to drought in response to Mg supply, depending on the stress level. At adequate Mg supply, M8808 plants were found to be the most sensitive genotype to water stress (Fig 3b and 3c), displaying low tolerance to moderate (42%) and severe (36%) drought (Fig 1b), whereas Bônus 8579 and TMG 1180 plants displayed elevated stress tolerance (Fig 3b and 3c), with similar

**Table 3. Mean comparison data for leaf and stem Mg, chlorophyll *a* (Chl *a*), *b* (Chl *b*) and *total* (Chl *total*), carotenoids, and ratios between Chl *a* and Chl *b* (Chl *a*/Chl *b*), and carotenoids and chlorophyll total (Car/Chl) from soybean genotypes (Bônus 8579, M8808 IPRO and TMG 1180) after 28 days grown under adequate and high magnesium regimes, and subjected to three drought treatments (control, moderate and severe drought).**

| Treatments | | | Leaf Mg | Stem Mg | Chl *a* | Chl *b* | Chl *total* | Carotenoids | Chl a/Chl b | Car/Chl |
|---|---|---|---|---|---|---|---|---|---|---|
| Stress level | Mg | Genotype | (g plant$^{-1}$) | (g plant$^{-1}$) | (µg plant$^{-1}$) | (µg plant$^{-1}$) | (µg plant$^{-1}$) | (µg plant$^{-1}$) | | |
| Control | Adequate | Bônus | 55.6±4.2$^{abA}$ | 42.6±1.3$^{bA}$ | 29.4±0.1$^{cA}$ | 9.0±0.1$^{cA}$ | 45.7±3.1$^{cA}$ | 5.8±0.1$^{bA}$ | 3.3±0.0$^{*aA}$ | 0.1±0.0$^{aB}$ |
| | | M8808 | 64.5±7.3$^{aA}$ | 78.2±4.1$^{*aA}$ | 61.2±1.0$^{*aA}$ | 19.7±0.2$^{*bA}$ | 96.8±6.7$^{*aA}$ | 10.3±0.8$^{*aA}$ | 3.1±0.0$^{aA}$ | 0.1±0.0$^{aA}$ |
| | | TMG | 40.8±2.0$^{bA}$ | 38.6±2.8$^{bA}$ | 47.8±5.2$^{bA}$ | 25.9±1.2$^{*aA}$ | 70.2±7.3$^{bA}$ | 7.7±0.8$^{abA}$ | 1.8±0.2$^{bA}$ | 0.1±0.0$^{aA}$ |
| | High | Bônus | 92.5±2.1$^{*aA}$ | 75.7±2.4$^{*aA}$ | 69.2±1.6$^{*aA}$ | 29.3±0.8$^{*aA}$ | 98.4±1.0$^{*aA}$ | 11.0±1.6$^{*aA}$ | 2.4±0.1$^{bA}$ | 0.1±0.0$^{aA}$ |
| | | M8808 | 63.5±0.1$^{1bA}$ | 50.8±4.3$^{bA}$ | 47.3±3.3$^{bA}$ | 14.3±0.4$^{cAB}$ | 61.0±5.2$^{bA}$ | 7.1±0.6$^{bA}$ | 3.3±0.2$^{*aA}$ | 0.1±0.0$^{aA}$ |
| | | TMG | 72.3±13.1$^{*bA}$ | 64.9±12.3$^{*abA}$ | 48.6±9.1$^{bA}$ | 20.2±2.4$^{bA}$ | 64.4±11.2$^{bA}$ | 6.7±1.3$^{bA}$ | 2.4±0.3$^{bB}$ | 0.1±0.0$^{aB}$ |
| Moderate drought | Adequate | Bônus | 33.1±1.1$^{aB}$ | 23.2±1.6$^{aB}$ | 25.9±0.6$^{aA}$ | 9.4±0.8$^{aA}$ | 23.2±0.8$^{aB}$ | 3.0±0.3$^{aB}$ | 2.8±0.3$^{*aA}$ | 0.1±0.0$^{aB}$ |
| | | M8808 | 24.5±0.7$^{aB}$ | 23.9±1.0$^{aB}$ | 21.9±0.8$^{aB}$ | 11.6±0.5$^{aB}$ | 38.0±1.7$^{aB}$ | 3.2±0.1$^{aB}$ | 1.9±0.1$^{bB}$ | 0.1±0.0$^{aA}$ |
| | | TMG | 23.7±0.9$^{aA}$ | 24.1±0.9$^{aA}$ | 25.4±0.5$^{aB}$ | 11.0±0.2$^{aB}$ | 37.1±0.6$^{aB}$ | 3.9±0.0$^{aB}$ | 2.3±0.1$^{abA}$ | 0.1±0.0$^{aA}$ |
| | High | Bônus | 43.3±0.8$^{bB}$ | 42.8±1.0$^{*aB}$ | 27.0±1.5$^{bB}$ | 13.3±1.0$^{bB}$ | 49.0±1.2$^{*bB}$ | 5.4±0.3$^{*bB}$ | 2.1±0.2$^{aA}$ | 0.1±0.0$^{aA}$ |
| | | M8808 | 69.6±0.5$^{*aA}$ | 58.7±0.6$^{*aA}$ | 50.5±2.6$^{*aA}$ | 18.5±2.0$^{*abA}$ | 65.9±1.7$^{*abA}$ | 8.4±0.5$^{*aA}$ | 2.8±0.2$^{*aA}$ | 0.1±0.0$^{*aA}$ |
| | | TMG | 64.2±2.0$^{*aA}$ | 55.9±3.6$^{*aAB}$ | 45.6±2.6$^{*aA}$ | 22.2±0.9$^{*aA}$ | 73.0±2.8$^{*aA}$ | 6.8±0.5$^{abA}$ | 2.1±0.1$^{aB}$ | 0.1±0.0$^{aB}$ |
| Severe drought | Adequate | Bônus | 38.6±0.9$^{aABB}$ | 22.0±1.1$^{aB}$ | 19.1±0.2$^{abA}$ | 7.1±0.3$^{abA}$ | 25.6±0.9$^{aABB}$ | 5.8±0.2$^{aA}$ | 2.7±0.1$^{abA}$ | 0.2±0.0$^{*aA}$ |
| | | M8808 | 32.0±3.2$^{aB}$ | 27.6±0.7$^{aB}$ | 12.1±0.6$^{bB}$ | 3.6±0.1$^{bC}$ | 17.3±0.6$^{aB}$ | 1.9±0.1$^{bB}$ | 3.3±0.2$^{aA}$ | 0.1±0.0$^{bA}$ |
| | | TMG | 40.4±2.6$^{aA}$ | 28.8±0.4$^{aA}$ | 26.5±0.9$^{aB}$ | 12.0±0.3$^{aB}$ | 38.0±1.3$^{aB}$ | 3.5±0.2$^{abB}$ | 2.2±0.0$^{bA}$ | 0.1±0.0$^{bA}$ |
| | High | Bônus | 35.2±1.1$^{bB}$ | 24.3±0.4$^{bC}$ | 28.3±0.7$^{aB}$ | 11.1±1.0$^{aB}$ | 40.1±3.1$^{aB}$ | 4.3±0.5$^{bB}$ | 2.6±0.2$^{aA}$ | 0.1±0.0$^{bA}$ |
| | | M8808 | 55.6±5.1$^{*aA}$ | 51.2±4.6$^{*aA}$ | 34.2±0.9$^{*aB}$ | 12.1±0.4$^{*aB}$ | 46.9±1.4$^{*aA}$ | 6.5±0.5$^{*abA}$ | 2.8±0.1$^{aA}$ | 0.1±0.0$^{bA}$ |
| | | TMG | 38.4±2.4$^{*abB}$ | 45.7±4.2$^{*aB}$ | 23.6±1.2$^{aB}$ | 8.0±0.5$^{aB}$ | 37.1±3.5$^{aB}$ | 7.2±0.2$^{bA}$ | 3.0±0.3$^{*aA}$ | 0.2±0.0$^{*aA}$ |

*Note*: Different capital letters indicate significant differences due to drought stress within the same genotype and Mg level. Different lowercase letters denote significant alterations among soybean genotypes within the same Mg and stress level. The presence of asterisks (*) represent significant differences due to Mg treatment in the same genotype and drought level, according to Tukey's test (p < 0.05).

relative indexes of approximately 62 and 54% for moderate and severe drought, respectively (Fig 1a and 1c). Nonetheless, M8808 genotype was the most responsive genotype to Mg supplementation under drought (Figs 2b, 3b and 3c), and high Mg-fed stressed plants almost recovered growth at both drought levels compared to the control (Fig 1b), displaying elevated tolerance in all water regimes. On the other hand, although Mg supplementation strongly increased plant growth under control conditions for TMG 1180 and Bônus 8579 plants (Fig 2a and 2c), it did not promote conspicuous increase in drought tolerance of these genotypes (Fig 2a and 2c). Thus, the high values of relative tolerance to moderate and severe drought from TMG 1180 plants (Fig 1c) seem to result from constitutive plasticity rather than Mg supplementation-stimulated tolerance (Fig 2c). Additionally, the Bônus 8579 genotype was found to be unresponsive to Mg supplementation (Fig 3b and 3c).

## Water status

No significant alteration was observed in the Ψs among soybean genotypes, irrespective of Mg supply and stress level (Fig 3 and Table 2); but Bônus 8579 plants showed decreased Ψs under moderate drought compared to the control and severe stress (Fig 1a). Additionally, under moderate drought, Mg-supplemented M8808 plants displayed lower Ψs than those from Mg-recommended ones (Fig 2b).

Relative water content (RWC) and leaf succulence (LS) were differentially regulated by the studied treatments. At adequate Mg, drought-stressed TMG1180 plants displayed increased RWC values at both stress levels compared to the control (Fig 1c). Conversely, Mg-

supplemented M8808 and Bônus 8579 plants exhibited a significant increase in RWC under moderate and severe drought, as related to the respective controls (Fig 1a and 1b); however, LS was dramatically decreased by drought imposition in M8808 and TMG 1180 plants (Fig 1b and 1c). In general, the highest values of RWC in moderately and severely stressed plants were observed in the M8808 genotype, in both adequate and supplementation Mg nutrition, in relation to Bônus 8579 and TMG 1180 plants (Fig 3). Little or no significant alteration was observed for LS among the studied soybean genotypes for all Mg and drought treatments (Fig 3 and Table 2).

## Mg accumulation in plant tissues

The Mg content in leaves and stems was drastically decreased by moderate and severe drought in Bônus 8579 plants (Fig 1a); whereas similar response for M8808 genotype was exhibited only under adequate Mg treatments (Fig 1b). Also, a significant decrease in Mg content under drought stress in the TMG genotype was detected exclusively in the leaves of severely stressed plants under Mg supplementation (Fig 1c and Table 3). Under moderate and severe drought, an increase in leaf and stem Mg accumulation was noticed only in Mg-supplemented M8808 plants compared to those treated with adequate Mg levels (Fig 2b). Thus, the most relevant alterations in Mg accumulation among soybean genotypes were noticed in plants under high Mg supply, highlighting that M8808 and TMG plants under moderate and severe drought demonstrate leaf and stem Mg contents greater than the Bônus 8579 genotype (Fig 3b and 3c). For all cases, the highest Mg accumulation was found in M8808 plants.

## Photosynthetic pigments

Moderate and severe drought promoted significant decreases in Chl *a*, Chl *b*, Chl *total*, and carotenoids of the Bônus 8579 genotype, exclusively in plants from Mg supplementation treatments (Fig 1a and Table 3). In contrast, for the M8808 genotype, significant reductions in all photosynthetic pigments by moderate and severe drought were noticed only in plants under adequate Mg treatments (Fig 1b). TMG 1180 plants also exhibited strong decreases in Chl *a*, Chl *b*, Chl *total*, and carotenoids under moderate drought under adequate Mg and significant decreases under severe drought in Chl *a*, Chl *b* and Chl *total* under Mg supplementation (Fig 1c). Under moderate drought, the soybean plants showed increase in all photosynthetic pigments by Mg supplementation, except for Chl *a* and Chl *b* of Bônus genotype (Fig 2); whereas a similar but more intense response was reported Mg-supplemented M8808 plants subjected to moderate and severe drought compared to those Mg-adequate ones (Fig 2b).

Taken together, under adequate Mg supply, the data comparison among soybean genotypes clearly demonstrates that unstressed M8808 plants showed higher photosynthetic pigments than the Bônus 8579 and TMG 1180 plants (Fig 3a), a response downregulated under moderate drought (Fig 3b) and dramatically decreased by severe stress (Fig 3c). Curiously, Mg supplementation did not promote increase in pigments of well-irrigated M8808 plants compared to other soybean genotypes (Fig 3a and Table 3), but it seemed to be crucial for increasing the contents of Chl *a*, Chl *b*, Chl *total*, and carotenoids from the M8808 genotype subjected to moderate (Fig 3b) and severe (Fig 3c) drought. Additionally, the Chl *a*/Chl *b* and Car/Chl *total* ratios Mg-supplemented M8808 plants under moderate drought were higher than in other soybean genotypes (Fig 3b). and the opposite was noticed under severe drought (Fig 3c).

## Principal component analysis (PCA)

PCA was performed to investigate the correlation between magnesium supply and parameters that best separated the soybean genotypes for tolerance to drought (Fig 4). The data explained

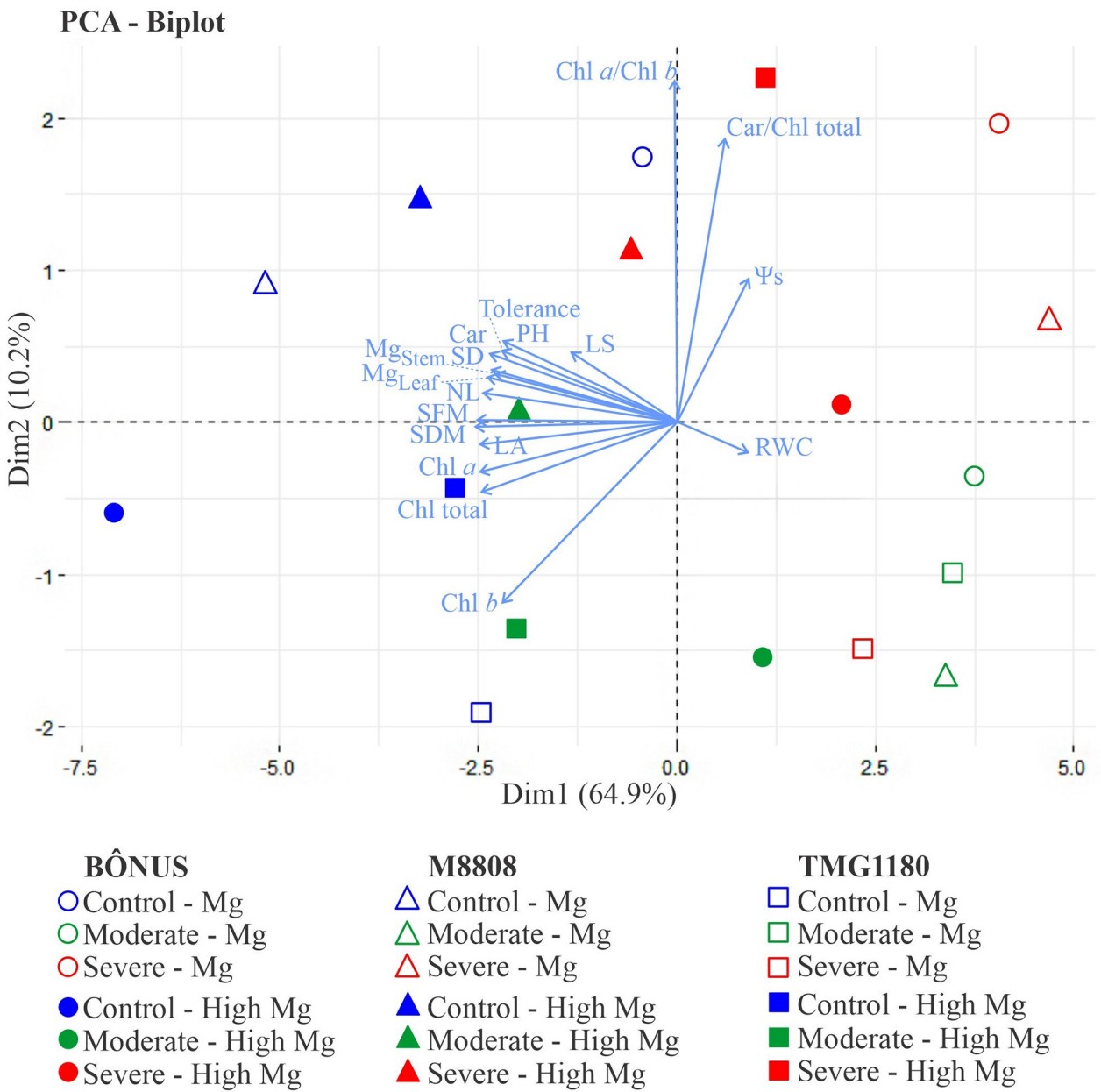

**Fig 4. Principal component analysis (PCA).** Scatter plots of the studied variables for Bônus 8579, M8808 IPRO and TMG 1180 soybean genotypes subjected to drought and Mg supply treatments. The percent variance for each PC is specified on axes X and Y. PCA shows the loading plot of variables: plant height (PH), stem diameter (SD), number of leaves (NL), leaf area (LA), shoot fresh mass (SFM), shoot dry mass (SDM), relative tolerance to drought (tolerance), osmotic potential ($\Psi$s), relative water content (RWC), leaf succulence (LS), chlorophyll $a$ (Chl $a$), chlorophyll $b$ (Chl $b$), chlorophyll *total* (Chl *total*), carotenoids (Car), Chl $a$/Chl $b$ ratio and Car/Chl *total* ratio.

78.51% of the total variation, with 68.15 and 10.36% explaining the first and second components, respectively. The biplot demonstrates an excellent separation between the groups based on adequate and supplementation Mg treatments (Fig 4); nevertheless, there was an overlap for soybean genotypes and drought levels, suggesting the existence of similar responses for groups. For all cases, the most expressive responses were registered in 1.7 Mg-supplemented

stressed M8808 plants under both moderate and severe drought, displaying a recovery performance close to that of the control treatments.

According to Pearson correlation coefficients, plant growth correlated significantly with photosynthetic pigments (Chl *a*, Chl *b*, Chl *total*, and carotenoids), indicating a positive correlation with well-irrigated plants (Fig 4). Relative tolerance to drought was closely related to leaf succulence, whereas relative water content correlated significantly with Ψs and the Chl *a*/Chl *b* and Car/Chl *total* ratios, showing a high positive correlation with moderately and severely stressed plants (Fig 4).

## Discussion

In a preliminary study under greenhouse conditions, we challenged seven soybean genotypes (AS3810 IPRO, M8644 IPRO, TMG 1180, NS8338 IPRO, BMX 81I81, M8808 IPRO and Bônus 8579) to several drought treatments to choose the most drought-contrasting genotypes. Based on plant growth and indicators of stress tolerance, we observed that M8808 is drought-sensitive; Bônus 8579 is tolerant to moderate drought and discreetly tolerant to severe stress; and TMG is highly tolerant to drought [27]. In the current study, field experiments were carried out to evaluate the role of Mg supplementation in triggering water stress acclimation in drought-sensitivity contrasting soybean genotypes. Concordantly, under the recommended Mg nutrition, the TMG 1180 and Bônus 8579 genotypes exhibited elevated tolerance to moderate and severe drought, whereas M8808 was found to be the most sensitive genotype (Fig 3b and 3c).

Mg treatments were able to modulate soybean responses in a genotype- and stress-level-dependent manner. High Mg supply (supplementation) promoted an increase in growth and photosynthetic pigments for all studied genotypes under well irrigation and moderate drought stress (Fig 2, Tables 1 and 3). Nonetheless, improved plant performance induced by Mg supplementation under severe drought was evident only in M8808 plants (Figs 2b and 3c), suggesting that Mg supply can recover the growth of a plant genotype recognized as sensitive to drought. Furthermore, photosynthetic pigment ratios were lower in M8808 plants under severe drought (Fig 3c), indicating a greater enhanced ability to capture and dissipate light [28]. In contrast, Bônus 8579 plants were found to be dramatically affected by severe drought, as proven by the increased Car/Chl *total* ratio (Figs 1a, 3b and 3c), suggesting an energy excess incoming in the electron transport chain [28].

Mg-induced drought tolerance in M8808 plants was associated with increased RWC in leaf tissues (Figs 1b, 2b, 3b and 3c), which has been considered a key factor for estimating drought tolerance in plants [29–31]. The elevated water relative content was most likely an attempt to counteract tissue dissection and maintain cell turgidity under low water availability once the leaf succulence in M8808 plants was slightly decreased under drought (Figs 1b and 2b). In addition, the results suggest that Mg supplementation-induced RWC regulation does not seem to result from osmotic adjustment under water stress, as the Ψs from M8808 plants remained unaltered or increased under drought (Figs 1b and 2b). Nonetheless, these Mg-supplemented plant groups showed the lowest Ψs values under severe drought compared to the other soybean genotypes (Fig 3c).

Elevated performance of drought-stressed M8808 plants under Mg supplementation was closely related to increased accumulation of photosynthetic pigments (Figs 2b, 3b, 3c and 4). The data suggest that high Mg availability in the medium increased its uptake by root cells, which was translocated to and accumulated in leaves (Figs 1b and 2b) and contributed to the photosynthetic process by promoting chlorophyll biosynthesis [32, 33]. Thus, the M8808 plants were able to restore growth even at low water availability in the soil (Fig 2b). The

differential response for chlorophyll accumulation among the soybean genotypes clearly demonstrates their genetic potential to cope with drought damages. Similar responses were also reported in wheat and maize plants under heat stress [20] and in *Torreya grandis* seedlings under heavy metal stress [34]. Furthermore, plants treated with Mg are more photosynthetically efficient and lose less water during the process [17].

The combined effects of Mg supply and drought level seem to be stress- and dose-dependent, since PCA segregated all treatments. Although drought stress promoted growth reduction in all soybean genotypes, Mg supplementation was found to counteract water deficit effects, particularly for M8808 plants. In this case, severely and moderately stressed M8808 plants fed the highest Mg dosage overcome drought damage and exhibited performance close to that of control plants (Fig 4). The interaction of one or two factors demonstrates the sensitivity of soybean to the studied treatments. This reprogramming suggests a different response mechanism for plants mediated by Mg supply and drought treatment. Certainly, Mg upregulates plant growth when plants are well irrigated, but this regulation is genotype-dependent in drought-stressed plants (Fig 4) [35].

## Conclusion

Our investigative study revealed that magnesium supplementation in soil improves tolerance to drought in M8808 soybean plants. The beneficial effects of a high Mg supply on increasing the tolerance of the M8808 genotype to water stress are associated with improved relative water content and Mg accumulation, which resulted in an alleviation of tissue dissection and an increased photosynthetic pigment content. Our findings indicate that Mg supplementation at 1.7 $cmol_c$ $dm^{-3}$ could be employed to prevent growth losses due to drought and may have a significant practical application as a potential technique for cultivating soybean plants in arid and/or semiarid regions.

## Supporting information

**S1 Fig. Environmental data dynamics.** Dynamics of mean, maximum, minimum temperature, relative humidity and evapotranspiration after sowing.
(PDF)

**S1 Table. Chemical analysis of soil before the trials.** Chemical analysis of soil before the experiment for growing soybean plants.
(PDF)

**S2 Table. Chemical analysis of soil after the trials.** Chemical analysis of soil after the experiment with soybean plants.
(PDF)

**S3 Table. Summary of variance analysis (ANOVA) of studied variables.** Split plot analysis of variance results for studied variables in soybean plants.
(PDF)

## Acknowledgments

The authors are grateful for farm Celeiro Sementes for kindly providing the seeds of the soybean genotypes.

## Author Contributions

**Conceptualization:** Amanda Soares Santos, Ramilos Rodrigues de Brito, Enéas Gomes-Filho, Marcos Renan Lima Leite, Alejandro Manuel Ecos Espino, Rafael de Souza Miranda.

**Data curation:** Amanda Soares Santos, Alana Cavalcante da Silva, Jennyfer Yara Nunes Batista, Bruno Sousa Figueiredo da Fonseca, Francisco de Alcântara Neto, Rafael de Souza Miranda.

**Formal analysis:** Davielson Silva Pinho, Alana Cavalcante da Silva, Stelamaris de Oliveira Paula-Marinho, Rafael de Souza Miranda.

**Funding acquisition:** Alan Mario Zuffo, Francisco de Alcântara Neto, Pedro Arias Cubillas.

**Investigation:** Amanda Soares Santos, Davielson Silva Pinho, Alana Cavalcante da Silva, Ramilos Rodrigues de Brito, Julian Junio de Jesus Lacerda, Everaldo Moreira da Silva, Jennyfer Yara Nunes Batista, Bruno Sousa Figueiredo da Fonseca, Jorge González Aguilera, Rafael de Souza Miranda.

**Methodology:** Julian Junio de Jesus Lacerda, Bruno Sousa Figueiredo da Fonseca, Rafael de Souza Miranda.

**Resources:** José Antonio Rodríguez García, Milko Raúl Rivera Campano.

**Supervision:** Ramilos Rodrigues de Brito.

**Validation:** Everaldo Moreira da Silva, Enéas Gomes-Filho, Stelamaris de Oliveira Paula-Marinho, Hebert Hernán Soto Gonzales.

**Visualization:** Julian Junio de Jesus Lacerda, Stelamaris de Oliveira Paula-Marinho.

**Writing – original draft:** Amanda Soares Santos, Rafael de Souza Miranda.

**Writing – review & editing:** Enéas Gomes-Filho, Alexson Filgueiras Dutra, Alan Mario Zuffo, Francisco de Alcântara Neto, Rafael de Souza Miranda.

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
