## [Decision Letter · Decision Letter 0]

6 Jun 2023

PONE-D-23-05527Magnesium supplementation alleviates drought damage during vegetative stage of soybean plantsPLOS ONE

Dear Dr. Miranda,

Thank you for submitting your manuscript to PLOS ONE. After careful consideration, we feel that it has merit but does not fully meet PLOS ONE’s publication criteria as it currently stands. Therefore, we invite you to submit a revised version of the manuscript that addresses the points raised during the review process.

We look forward to receiving your revised manuscript.

Kind regards,

Ramachandran Srinivasan, Ph.D.

Academic Editor

PLOS ONE

Journal Requirements:

"Fundação de Amparo à Pesquisa do Estado do Piauí (FAPEPI) under grant Edital PPP FAPEPI/MCT/CNPq/CT-INFRA n° 007/2018.

Conselho Nacional de Desenvolvimento Científico e Tecnológico (CNPq) under grant number 427219/2018-3."

"The authors are grateful for fellowship provided by Coodenação de Aperfeiçoamento de Pessoal de Nível Superior (CAPES) for A.S. Santos and for financial support provided by the Fundação de Amparo à Pesquisa do Estado do Piauí (FAPEPI) under grant Edital PPP FAPEPI/MCT/CNPq/CT-INFRA n° 007/2018 and Conselho Nacional de Desenvolvimento Científico e Tecnológico (CNPq) under grant number 427219/2018-3.."

"Fundação de Amparo à Pesquisa do Estado do Piauí (FAPEPI) under grant Edital PPP FAPEPI/MCT/CNPq/CT-INFRA n° 007/2018.

Conselho Nacional de Desenvolvimento Científico e Tecnológico (CNPq) under grant number 427219/2018-3."

6. Thank you for stating the following in your Competing Interests section: "The authors declare that they have no conflicts of interest."

7. We note that you have referenced (ie. Unpublished [page 10]) which has currently not yet been accepted for publication. Please remove this from your References and amend this to state in the body of your manuscript: (ie “Bewick et al. [Unpublished]”) as detailed online in our guide for authors

**Additional Editor Comments:**

Dear Author,

Carefully, revise the manuscript as suggested by Reviewer 1 and 2 and respond to reviewer queries in point by point separately as Response to reviewer

Thanking you.

Reviewers' comments:

Reviewer's Responses to Questions

**Comments to the Author**

1. Is the manuscript technically sound, and do the data support the conclusions?

Reviewer #1: Yes

Reviewer #2: Yes

2. Has the statistical analysis been performed appropriately and rigorously? 

Reviewer #1: I Don't Know

Reviewer #2: Yes

3. Have the authors made all data underlying the findings in their manuscript fully available?

Reviewer #1: No

Reviewer #2: Yes

4. Is the manuscript presented in an intelligible fashion and written in standard English?

Reviewer #1: Yes

Reviewer #2: No

5. Review Comments to the Author

Reviewer #1: The current manuscript "Magnesium supplementation alleviates drought damage during vegetative stage of soybean plants" is an interesting study. I have a few recommendations which I believe might help to improve the current status of the manuscript and request the authors to consider them.

1. There are a few spelling errors in the manuscript. For ex. line number 84 "bioEstimulants" instead of "biostimulants". Please check the manuscript tp eliminate such errors.

2. I feel that the study would be strong if the introduction part provides some statistical information on the actual losses in soybean due to drought. Since the preliminary and current experiments are meticulous, it would be meaningful if the severity of drought stress particularly in soybean is explained in a real world scenario.

3. I am a bit confused with the data presented in Figures 1-3. I believe they are from the same dataset but presented in different aspects. Would just one figure not be enough to explain? I am sorry, I am unable to follow the authors point of view on the this.

I am also unable to find the tables, which I am not sure of the reason for this.

Reviewer #2: The authors present interesting data in "Magnesium supplementation alleviates drought damage during vegetative stage of

soybean plant". Having said this, the current version of this manuscript is far from meeting the basic requirements, both in terms of form or scientific rigor in writing, of a scientific publication. At many times statements in the text are not supported by references, many reference are missing from the reference list, many faults in the way the Methods are reported, lacking information on the experiments conducted, lacking information on the Figures and the failure to relate these data to similar observation in the Discussion section.

In particular the issues to be addressed are:

1. Please check the sentence in Line 61 & 62 which is not clear.

2) Rewrite the sentence in Line 87 & 88, "especially through activation of mechanisms for water retention in plant tissues" not clear. Provide a clear explanation.

3) Why the author have chosen drought treatments to be studied 28 days after sowing?

4) Are the results are applicable to the other plant species?

5) Do the authors have quantified the rate of protein synthesis? Because Mg influence the protein synthesis rate.

6) Have the authors carried out the experiment for root biomass?

7) Enzymes like ribulose-1,5-bisphosphate carboxylase/oxygenase (Rubisco), protein kinases, RNA

polymerase, glutathione synthase, adenosine triphosphatases (ATPases), phosphatases, and

carboxylases require Mg for activation. Does the author have observed any excess load of Mg, if so does it exhibited any negative impact ?

6. PLOS authors have the option to publish the peer review history of their article (what does this mean?). If published, this will include your full peer review and any attached files.

Reviewer #1: **Yes: **Parthiban Subramanian

Reviewer #2: No

---

## [Author Response · Author response to Decision Letter 0]

20 Jun 2023

We would like to thank the Reviewers and Editorial team for their comments that helped to improve the clarity and quality of the manuscript. All comments were considered, and the alterations were provided in the text tracked-changes. Once the text was altered, a proofreading was performed to meet all requirements appropriately. Please see the responses to reviewers’ comments below.

Journal Requirements:

R1: The paper was completely revised to attend the PLOS ONE’s style requirements.

R2: We want to comment that no permission was required to access the experimental site. Thus, a brief statement was added in Methods section in order to meet the Journal Requirements.

R3: The information was revised and provided appropriately. 

R4: We add the requested information in the cover letter in order to attend the Journal Requirements.

R5: We add the requested information in the cover letter in order to attend the Journal Requirements.

6. The information about Competing Interests should be included in your cover letter; we will change the online submission form on your behalf.

R6: We add the requested information in the cover letter in order to attend the Journal Requirements.

7. We note that you have referenced (ie. Unpublished [page 10]) which has currently not yet been accepted for publication. Please remove this from your References and amend this to state in the body of your manuscript: (ie“Bewick et al. [Unpublished]”) as detailed online in our guide for authors

R7: We change the reference for Miranda et al. [Unpublished] as suggested.

8. Please include captions for your Supporting Information files at the end of your manuscript, and update any in-text citations to match accordingly.

R8: The captions of Supporting Information were added at the end of manuscript and all citations were carefully revised in the text.

Reviewer #1

1. “There are a few spelling errors in the manuscript. For ex. line number 84 "bioEstimulants" instead of "biostimulants". Please check the manuscript to eliminate such errors.”

R9: We apologize for the mistakes. The manuscript was completely revised in order to remediate these errors. We thank for comments. 

2. “I feel that the study would be strong if the introduction part provides some statistical information on the actual losses in soybean due to drought. Since the preliminary and current experiments are meticulous, it would be meaningful if the severity of drought stress particularly in soybean is explained in a real world scenario.”

R10: We revise all introduction and include the requested information in order to attend the Reviewer suggestion.

3. The Reviewer said: “I am a bit confused with the data presented in Figures 1-3. I believe they are from the same dataset but presented in different aspects. Would just one figure not be enough to explain? I am sorry, I am unable to follow the authors point of view on the this.”

R11: We understand the reviewer's doubt. We would like to comment that, as the research was carried out in a triple factorial, we have decided to present the data in an integrated view to facilitate the reader's understanding. Each figure (1, 2, and 3) presents a cluster analysis with mean comparison for one independent factor, using a reference for each figure. Figure 1 shows the results of drought treatments comparison within Mg levels and soybean genotypes using well-irrigated plants as reference. The figure 2 presents the results of Mg treatments comparison within drought levels and soybean genotypes using plants grown under recommended Mg as reference. Yet, figure 3 shows a comparison of soybean genotypes within Mg and drought treatments using the data from Bônus genotype as reference. All of these data were incorporated into tables 1, 2, and 3, including the absolute values and statistical results. Anyway, in order to clarify the Reviewer doubt, we have made the following changes in the current version of the paper: i) added an additional title to the caption of each figure; ii) included the ANOVA table as supplementary material (Supplementary S3 Table); and iii) inserted a paragraph (see lines 159 to 167 – clean version) in the results section explaining the approach used for presenting the results. However, we remain open to any further suggestions from the reviewers.

Reviewer #2

The Reviewer said “Having said this, the current version of this manuscript is far from meeting the basic requirements, both in terms of form or scientific rigor in writing, of a scientific publication. At many times statements in the text are not supported by references, many reference are missing from the reference list, many faults in the way the Methods are reported, lacking information on the experiments conducted, lacking information on the Figures and the failure to relate these data to similar observation in the Discussion section.”

R12: We do not understand this general comment. Anyway, we carefully revised all manuscript in order to find the raised mistakes (missing reference and information). Please see the alterations in the text. 

1). “Please check the sentence in Line 61 & 62 which is not clear.”

R13: This sentence and other portions of introduction were rewritten in order to attend the Reviewer suggestion. Please, see the paper with tracked changes.

2) “Rewrite the sentence in Line 87 & 88, "especially through activation of mechanisms for water retention in plant tissues" not clear. Provide a clear explanation”

R14: The sentence was rewritten in order to attend the Reviewer suggestion: Our working hypothesis is that Mg supplementation triggers water stress tolerance in G. max through activation of mechanisms for maintenance of water content in plant tissues.

3) “Why the author have chosen drought treatments to be studied 28 days after sowing?”

R15: In the current study, we try to investigate the implications of Mg supplementation during the vegetative stage of soybean plants. The time for drought imposition was defined to start at the V4 stage, based on observations from preliminary studies. In this case, the soybean plants in the field reached the V4 stage 28 days after sowing. Additionally, the sampling time was defined to be at the end of the vegetative stage, specifically at the beginning of the flowering stage (R1 stage), which coincided with 28 days after drought imposition (or 56 days after sowing). To address this uncertainty, we have revised certain sentences in the Materials and Methods section:

Lines 101-101 (clean version): Drought treatments were applied soon after the soybean plants achieved the V4 stage, 28 days after sowing.

Lines 117-118 (clean version): The plant material was harvested at 28 days after the imposition of drought treatments, before the reproductive stage.

4) “Are the results are applicable to the other plant species?”

R16: We have understood the reviewer's question. The supplementation of Mg has been proven to be beneficial, depending on the genotype and species, as stated in our study. Therefore, the application of these results to other plant species should be approached with caution and based on scientific investigations. For instance, we have already completed trials using Mg supplementation for cowpea in both greenhouse and field environments, and the data are encouraging, highlighting its role in mitigating drought damage and enhancing stress tolerance, particularly in sensitive genotypes. Surprisingly, it appears that genotypes of plant species such as soybean and cowpea, which have intrinsic tolerance, do not respond to Mg treatments under drought conditions. Considering this, we can recommend Mg supplementation for cowpea and soybean under water restrictions, but we cannot recommend it for other crop species without conducting scientific investigations.

5) “Do the authors have quantified the rate of protein synthesis? Because Mg influence the protein synthesis rate.”

R17: In the present study, our focus was to investigate the influence of Mg supplementation on the activation of drought tolerance, analyzing growth and key tolerance indicators. We would like to inform that we are just carrying out a new study to address biochemical investigations, including protein quantification, antioxidant enzymes, solute accumulation, and plant metabolome profiling. We understand the reviewer's comments, and while these suggestions will be incorporated into a future study.

6) “Have the authors carried out the experiment for root biomass?”

R18: We appreciate the reviewer's question. In this study, we did not include an analysis of root biomass. In fact, we had harvested the roots from the experiments, but we decided to exclude this assay from the paper due to concerns regarding its reliability. By considering a field trial, during the process of harvesting the roots, a portion of them remained in the soil (due to breakage) and could not be fully collected, especially in the water deficit treatments. Additionally, some roots from neighboring plants were in contact and intertwined with each other. These factors collectively raised doubts about the reliability of the root biomass data, leading us to choose not to include it in the paper.

7) “Enzymes like ribulose-1,5-bisphosphate carboxylase/oxygenase (Rubisco), protein kinases, RNA polymerase, glutathione synthase, adenosine triphosphatases (ATPases), phosphatases, and

carboxylases require Mg for activation. Does the author have observed any excess load of Mg, if so does it exhibited any negative impact?”

R19: We appreciate the reviewer's comments and inquiries. Indeed, there was a higher accumulation of Mg in the plants supplemented with Mg, particularly observed in the M8808 genotype when subjected to drought. As mentioned in response R17, we did not include biochemical assays in the present article. We thank you for your comment, and these investigations will be taken into consideration for the new study that is currently underway.

---

## [Decision Letter · Decision Letter 1]

10 Jul 2023

Magnesium supplementation alleviates drought damage during vegetative stage of soybean plants

PONE-D-23-05527R1

Dear Dr. Rafael de Souza Miranda,

We’re pleased to inform you that your manuscript has been judged scientifically suitable for publication and will be formally accepted for publication once it meets all outstanding technical requirements.

Kind regards,

Ramachandran Srinivasan, Ph.D.

Academic Editor

PLOS ONE

Additional Editor Comments (optional):

Reviewers' comments:

Reviewer's Responses to Questions

**Comments to the Author**

1. If the authors have adequately addressed your comments raised in a previous round of review and you feel that this manuscript is now acceptable for publication, you may indicate that here to bypass the “Comments to the Author” section, enter your conflict of interest statement in the “Confidential to Editor” section, and submit your "Accept" recommendation.

Reviewer #1: All comments have been addressed

Reviewer #2: All comments have been addressed

2. Is the manuscript technically sound, and do the data support the conclusions?

Reviewer #1: Yes

Reviewer #2: Yes

3. Has the statistical analysis been performed appropriately and rigorously? 

Reviewer #1: Yes

Reviewer #2: Yes

4. Have the authors made all data underlying the findings in their manuscript fully available?

Reviewer #1: Yes

Reviewer #2: Yes

5. Is the manuscript presented in an intelligible fashion and written in standard English?

Reviewer #1: Yes

Reviewer #2: Yes

6. Review Comments to the Author

Reviewer #1: (No Response)

Reviewer #2: (No Response)

7. PLOS authors have the option to publish the peer review history of their article (what does this mean?). If published, this will include your full peer review and any attached files.

Reviewer #1: **Yes: **Parthiban Subramanian

Reviewer #2: **Yes: **Jeyapragash Danaraj

---

## [Editor Report · Acceptance letter]

13 Jul 2023

PONE-D-23-05527R1 

Magnesium supplementation alleviates drought damage during vegetative stage of soybean plants 

Dear Dr. Miranda:

I'm pleased to inform you that your manuscript has been deemed suitable for publication in PLOS ONE. Congratulations! Your manuscript is now with our production department. 

Kind regards, 

on behalf of

Dr. Ramachandran Srinivasan 

Academic Editor

PLOS ONE